# Refining Genotypes and Phenotypes in *KCNA2*-Related Neurological Disorders

**DOI:** 10.3390/ijms22062824

**Published:** 2021-03-10

**Authors:** Jan H. Döring, Julian Schröter, Jerome Jüngling, Saskia Biskup, Kerstin A. Klotz, Thomas Bast, Tobias Dietel, G. Christoph Korenke, Sophie Christoph, Heiko Brennenstuhl, Guido Rubboli, Rikke S. Møller, Gaetan Lesca, Yves Chaix, Stefan Kölker, Georg F. Hoffmann, Johannes R. Lemke, Steffen Syrbe

**Affiliations:** 1Division of Paediatric Epileptology, Centre for Paediatric and Adolescent Medicine, University Hospital Heidelberg, 69120 Heidelberg, Germany; jan.doering@uni-heidelberg.de (J.H.D.); julian.schroeter@med.uni-heidelberg.de (J.S.); 2Praxis für Humangenetik Tübingen, 72076 Tuebingen, Germany; jerome.juengling@humangenetik-tuebingen.de (J.J.); saskia.biskup@humangenetik-tuebingen.de (S.B.); 3CEGAT GmbH, 72076 Tuebingen, Germany; 4Department of Neuropediatrics and Muscle Disorders, Medical Center, Faculty of Medicine, University of Freiburg, 79106 Freiburg, Germany; kerstin.alexandra.klotz@uniklinik-freiburg.de; 5Department of Epileptology, Medical Center, Faculty of Medicine, University of Freiburg, 79085 Freiburg, Germany; 6Medical Faculty, University Hospital Freiburg, 79085 Freiburg, Germany; tbast@diakonie-kork.de; 7Epilepsy Center Kork, Medical Faculty of the University of Freiburg, 77694 Kehl, Germany; tdietel@diakonie-kork.de; 8Department of Neuropediatrics, University Children’s Hospital, Klinikum Oldenburg, 26133 Oldenburg, Germany; korenke.georg-christoph@klinikum-oldenburg.de (G.C.K.); christoph.sophie@klinikum-oldenburg.de (S.C.); 9Division of Paediatric Neurology and Metabolic Medicine, Centre for Paediatric and Adolescent Medicine, University Hospital, 69120 Heidelberg, Germany; heiko.brennenstuhl@med.uni-heidelberg.de (H.B.); Stefan.Koelker@med.uni-heidelberg.de (S.K.); Georg.Hoffmann@med.uni-heidelberg.de (G.F.H.); 10Danish Epilepsy Centre, 4293 Dianalund, Denmark; guru@filadelfia.dk (G.R.); rimo@filadelfia.dk (R.S.M.); 11Department of Regional Health Research, University of Southern Denmark, 5230 Odense, Denmark; 12Department of Medical Genetics, Lyon University Hospital, University of Lyon, 69100 Lyon, France; gaetan.lesca@chu-lyon.fr; 13Pediatric Neurology Unit, Children’s Hospital, Purpan University Hospital, 31300 Toulouse, France; CHAIX.Y@chu-toulouse.fr; 14Institute of Human Genetics, University of Leipzig Medical Center, 04103 Leipzig, Germany; Johannes.Lemke@medizin.uni-leipzig.de; 15Center for Rare Diseases, University of Leipzig Medical Center, 04103 Leipzig, Germany

**Keywords:** voltage-gated potassium channel, K_v_1.2, shaker, epileptic encephalopathy, epilepsy, ataxia

## Abstract

Pathogenic variants in *KCNA2*, encoding for the voltage-gated potassium channel K_v_1.2, have been identified as the cause for an evolving spectrum of neurological disorders. Affected individuals show early-onset developmental and epileptic encephalopathy, intellectual disability, and movement disorders resulting from cerebellar dysfunction. In addition, individuals with a milder course of epilepsy, complicated hereditary spastic paraplegia, and episodic ataxia have been reported. By analyzing phenotypic, functional, and genetic data from published reports and novel cases, we refine and further delineate phenotypic as well as functional subgroups of *KCNA2*-associated disorders. Carriers of variants, leading to complex and mixed channel dysfunction that are associated with a gain- and loss-of-potassium conductance, more often show early developmental abnormalities and an earlier onset of epilepsy compared to individuals with variants resulting in loss- or gain-of-function. We describe seven additional individuals harboring three known and the novel *KCNA2* variants p.(Pro407Ala) and p.(Tyr417Cys). The location of variants reported here highlights the importance of the proline(405)–valine(406)–proline(407) (PVP) motif in transmembrane domain S6 as a mutational hotspot. A novel case of self-limited infantile seizures suggests a continuous clinical spectrum of *KCNA2*-related disorders. Our study provides further insights into the clinical spectrum, genotype–phenotype correlation, variability, and predicted functional impact of *KCNA2* variants.

## 1. Introduction

Potassium channels are involved in many vital processes such as neuronal excitability, neurotransmitter release, and cellular osmoregulation [1,2]. *KCNA2*, encoding for the voltage-gated potassium channel subfamily A member 2 (K_v_1.2, OMIM * 176262) is a member of the shaker-like delayed rectifier potassium channel family [3]. It is predominantly expressed in axons and presynaptic terminals of the central nervous system [4,5]. Homo- or heterotetramers formed by α-subunits containing variable proportions of K_v_-subfamily members contribute to functional channel characteristics [6]. K_v_1.2 consists of six transmembrane segments (S1–S6) forming the voltage-sensing domain (S4) and the ion-conducting pore (S5-S6) with its selectivity filter localized in the pore loop [7,8]. The C-terminal and N-terminal cytosolic domains are less conserved and regulate tetramerization and the association to modifying subunits [9,10]. Several antibody-mediated neurological diseases in children and adults have been linked to proteins associated with K_v_1 channel components, such as leucine-rich glioma-inactivated 1, contactin-associated protein 2, and particularly K_v_1 channel subunits [11,12].

In 2015, pathogenic de novo variants in *KCNA2* were first described in six individuals with early-onset severe epilepsy, cognitive impairment, and ataxia [13]. Two phenotypically distinct epilepsy syndromes were delineated correlating with gain-of-function (GOF) and loss-of-function (LOF) changes of the net potassium current of K_v_1.2 [13,14]. The recurrent *KCNA2* variant c.881G > A, leading to an amino-acid exchange at position 294 within the voltage sensing domain, was identified in individuals with hereditary spastic paraplegia (HSP) as an additional phenotype [15]. In 2017, novel and previously reported individuals were summarized into three distinct functional and clinical subgroups of K_v_1.2 dysfunction: GOF, LOF, and a mixed dysfunctional type (GOF/LOF) [16].

In this study, we present additional individuals with known and novel variants in *KCNA2*. We reevaluate phenotypic, functional, and genetic data from all published cases to refine and further delineate the assumed phenotypic and functional subgroups. We analyze the genetic distribution of all known pathogenic and likely pathogenic variants in *KCNA2* and predict the functional impact of all possible missense variations in *KCNA2*.

## 2. Results

We identified 115 individuals with pathogenic and likely pathogenic (P/LP) *KCNA2* variants from 28 publications (78 cases) and ClinVar database entries (37 cases). Additionally, we report clinical data from seven novel cases, six individuals with P and LP *KCNA2* variants, and one individual with a VUS. After the exclusion of all individuals without sufficient clinical information, we included 76 of 115 cases into further analyses (see Section 4).

In 40 individuals, variants occurred de novo, while 17 segregated with the disease in the family. In 21 cases, information on inheritance was unavailable. Thirty different alterations of *KCNA2* are on record to date, with 26 single-nucleotide variants (SNV), one in-frame deletion, and three different deletions (Table 1 and Table 2).

**Table 1 ijms-22-02824-t001:** *KCNA2* variants from published clinical reports.

cDNA Change	Protein Change	Functional Consequence	Phenotype	Number of Patients	Reference
**heterozygous variants:**					
**missense**					
c.469G > A	p.(Glu157Lys)	GOF	epileptic encephalopathy	1	[16]
c.676G > A	p.(Glu226Lys)	no data	epilepsy, childhood-onset	1	[17]
c.788T > C	p.(Ile263Thr)	LOF	epileptic encephalopathy	1	[13,16]
c.869T > G	p.(Leu290Arg)	GOF/LOF	epileptic encephalopathy	2	[16,18,19]
c.878T > A	p.(Leu293His)	GOF/LOF	epileptic encephalopathy	1	[16]
c.881G > A	p.(Arg294His)	LOF	hereditary spastic paraplegia	7	[15,20]
c.889C > T	p.(Arg297Trp)	no data	developmental and epileptic encephalopathy	3	[21,22,23]
c.890G > A	p.(Arg297Gln)	GOF	ataxia & myoclonic epilepsy	15	[3,13,14,16,24,25,26,27,28,29]
c.894G > T	p.(Leu298Phe)	GOF	epileptic encephalopathy	1	[13,16]
c.959C > T	p.(Thr320Ile)	no data	epilepsy, mild ataxia	1	[30]
c.971G > C	p.(Ser324Thr)	no data	epilepsy, drug-resistant	1	[31]
c.982T > G	p.(Leu328Val)	GOF/LOF	epileptic encephalopathy	2	[16,32]
c.1070G > A	p.(Gln357Arg)	no data	Lennox-Gastaut syndrome	1	[25]
c.1120A > G	p.(Thr374Ala)	GOF/LOF	epileptic encephalopathy, early onset	8	[16,26,33,34,35]
c.1192G > T	p.(Gly398Cys)	LOF	epileptic encephalopathy	1	[16]
c.1195G > A	p.(Val399Met)	no data	epileptic encephalopathy	1	[23]
c.1202C > T	p.(Thr401Ile)	no data	epileptic encephalopathy	1	[36]
c.1214C > T	p.(Pro405Leu)	LOF	epileptic encephalopathy	14	[13,16,23,34,37,38]
c.1223T > C	p.(Val408Ala)	no data	Rett-like syndrome with infantile onset seizures	1	[39]
**truncating**					
c.637C > T	p.(Gln213*)	LOF	epileptic encephalopathy	1	[16]
c.193C > T	p.(Arg65*)	predicted LOF	epilepsy	1	[17]
c.1265_1266delAG	p.(Glu422Glyfs*21)	predicted LOF	epileptic encephalopathy	1	[26]
**in-frame deletion**					
c.765_773del	p.(Met255_Ile257del)	LOF	episodic ataxia & pharmacoresponsive epilepsy	7	[3]
**deletion**					
c.110606081_111393713del~788 kb		no data	generalized epilepsy	1	[40]
**homozygous variants:**					
c.193C > T	p.(Arg65*)	no data	intellectual disability, autosomal-recessive	4	[41]

GOF: gain-of-function effects, LOF: loss-of-function effects compared to wild-type channels; all variants are described according to the transcript NM_004974.3.

### 2.1. Novel Cases with KCNA2 Variants in this Study

Data from seven previously unpublished cases with known pathogenic (*n* = 5) and novel variants (*n* = 2) in *KCNA2* were ascertained. Genetic and phenotypic data are summarized in Table 3. The localization and structural consequence of the novel missense variant p.(Tyr417Cys) is depicted in Figure 1. Furthermore a comparison of the wild-type proline(405)–valine(406)–proline(407) (PVP) motif and changes by the novel variant p.(Pro407Ala) andthe known variant p.(Pro405Leu) are shown. 

### 2.2. Clinical Characteristics of Individuals with Reported and Novel KCNA2 Variants

Mean age at inclusion and publication was 17 years (median age 10 years, range 0–81 years). No individual had died until the last reported study visit. Nearly all (91%) reported cases developed seizures. Mean age at seizure onset was 13 months (median age 8 months, range 0–156 months). In total, 41 male and 33 female individuals were included (three cases with missing information). No relevant differences in gender distribution on clinical manifestation or genotype were observed. Seven individuals were reported as not having seizures at all, of whom five harbor the variant p.(Arg294His) associated with hereditary spastic paraplegia and ataxia (HSP) [15,20] and two harbor the homozygous variant p.(Arg65*) associated with familial intellectual disability [41]. Associated movement and motor disorders were reported in 61/64 cases (95%), with ataxia in 64% of reported cases, tremor in 23%, and increased muscle tone (spasticity), in 16%. Febrile seizures were described in 28% (21/76) of cases. 30% (21/69) of the patients achieved seizure control (missing data in 11 cases). Of the cases with available data, most individuals (82%, 59/72) show intellectual disability of various severity. Thirteen individuals were reported with a normal cognitive development, including all five cases harboring the p.(Met255_Ile257del) deletion [3]. Cranial magnetic resonance imaging (available in 58 cases) was abnormal in 23 cases (40%), with cerebellar atrophy being most commonly reported (*n* = 18).

### 2.3. Phenotypic Features of Reported and Novel Patients, Grouped According to Known Functional Consequences on Kv1.2

Of all reported alterations of *KCNA2*, electrophysiological channel characterization has been available in 13/30 (43%) variants. Six different electrophysiological LOF variants in *KCNA2* have been described [3,13,15,16,20], three different *KCNA2* variants caused GOF in cellular expression systems [3,13,14,16,24,25,26] and four variants were reported having mixed functional effects (GOF/LOF) on mutated channels [16,19,34,35]. Of all reported and novel individuals, 61/76 (80%) harbored variants that have been electrophysiologically characterized. As proposed before, we grouped patients according to these known functional consequences on K_v_1.2 for phenotypic comparison (Table 4).

Compared to individuals with a mixed dysfunctional type (LOF/GOF), individuals with LOF variants showed better development before seizure onset (*p* = 0.012), later onset of epilepsy (*p* = 0.033), more favorable epilepsy outcome (seizure-free) (*p* = 0.014), and less severe intellectual disability (*p* = 0.029). Compared to individuals with LOF/GOF variants, epilepsy onset is significantly later in GOF individuals (*p* = 0.008), whereas development before seizures, epilepsy outcome, and intellectual disability did not differ significantly. Comparing individuals with LOF and GOF variants, development before seizures, epilepsy onset, seizure outcome, and intellectual disability was not significantly different. Figure 2 illustrates the age at onset of epilepsy in the different subgroups.

### 2.4. Prediction of Functional Relevance of KCNA2 Variants

To evaluate the likelihood of pathogenic variant effects, we combined the distribution of known pathogenic and benign variants with computational pathogenicity scores for all biologically possible missense variants of *KCNA2*. The REVEL meta-score combines variant annotations of 13 individual pathogenicity scores and showed high overall performance in the discrimination of pathogenic and benign variants in large clinical datasets [42,43]. High REVEL scores overlap with areas of pathogenic variant enrichment in *KCNA2* (Figure 3). Several peaks, corresponding to the voltage sensing domain (S4), the pore-loop (S5-S6) and the ion conducting pore with the gating mechanism (S5 and S6), were identified and are supported by similar tendencies in the evaluation of further established pathogenicity scores, such as CADD, M-CAP, PolyPhen, and SIFT (Appendix A). The density of putatively benign variants from the gnomAD database (green) shows enrichment at the less conserved N- and C-termini. In one specific region within S5, putatively benign variants are enriched, mainly being the result of tolerated changes at position Ala329.

## 3. Discussion

Recently, different neurological disorders, including epilepsy syndromes, ataxia, and hereditary spastic paraplegia were associated to K_v_1.2 channel dysfunction. In this study, we present additional patients with known and novel *KCNA2* variants. By analyzing phenotypic, functional, and genetic data from all published reports and novel cases, we refine and further delineate phenotypic and functional subgroups. Overall, the most prevalent manifestation was early-onset developmental and epileptic encephalopathy in combination with inconstant movement and motor disorders, mainly ataxia. Similar to other disorders related to voltage-gated ion channel genes (e.g., *KCNA1*, *KCNQ2*), a continuous spectrum from mild to severe neurologic manifestations and paroxysmal movement disorders can be delineated [44,45]. By comparing phenotypic characteristics from patients with functional data on mutational effects, carriers of variants leading to complex and mixed channel dysfunction that are associated with a gain and loss of potassium conductance more often show early developmental abnormalities and an earlier onset of epilepsy compared to patients with pure loss-of-function or pure gain-of-function variants, respectively.

### 3.1. Novel Cases of KCNA2 Variants and Clinical Presentation

We here report seven children with three known pathogenic variants and two novel variants in *KCNA2*. The variant p.(Tyr417Cys) was identified in a three-year-old boy (#1) with benign epilepsy of infancy (BFIS) and normal development. No other variants in known genes for BFIS were identified. The variant was inherited from an unaffected father. It is classified as a variant of unknown significance (VUS) according to the American College of Medical Genetics and Genomics (ACMG) (PM2, PP2, PP3) [46]. This variant is located at the cytosolic border of S6 in a region enriched for pathogenic variation and is predicted as damaging (Figure 3D, Appendix A). The amino acid Tyr417 is spared from variation in the gnomAD database. Previous functional investigations of the homologous amino acid exchange in drosophila p.(Tyr485Cys) have shown that cysteine replacement of tyrosine at this position leads to a slight reduction of single channel conductance (γ), an increased deactivation rate, and a small decrease in the time constant for “ON gating currents” [47,48]. The clinical phenotype with self-limiting infantile-onset seizures in this boy is partially overlapping with the mild end of *KCNA2*-related epilepsy. Therefore, a role of the specific *KCNA2* variant p.Tyr417Cys in benign infantile epilepsy (BFIS) appears possible, with a reduced penetrance, similar to other known genetic causes of infantile seizures such as *PRRT2* [49], *SCN2A* [50], and *SCN8A* [51]. The novel variant p.(Pro407Ala) was found in a three-year-old girl (#2) with neonatal-onset developmental and epileptic encephalopathy and profound intellectual disability (ID). The variant occurred de novo and has not been described before. It is absent from controls in the gnomAD database and predicted as damaging. According to ACMG criteria, this variant is likely pathogenic (PM6, PM2, PP2, PP3) [46]. The variant leads to an exchange from proline to alanine at position 407 within a region that is enriched for pathogenic variants. It disrupts the highly conserved proline(405)–valine(406)–proline(407) (PVP) motif, which is essential for channel gating. Variants affecting prolines in the PVP motif and in closely related residues have been identified in *KCNA2*- and recently in *KCNA1*-associated early-onset epileptic encephalopathy [16,23,36,39,52]. Previous in vitro data of p.(Pro407Ala) in the Shaker K_v_-channel suggested a non-conducting mutant channel as the underlying dysfunction [53]. This novel variant underlines the importance of the PVP motif regarding channel function.

The variant p.(Pro405Leu) affects the first proline of the PVP motif. We report two additional children with infantile-onset epilepsy and prolonged febrile seizures, harboring this recurrent variant. One of the two children had moderate intellectual disability, while the other one was described as normally developed. The mean age at seizure onset in all reported individuals with p.(Pro405Leu) was 9 months, and the main seizure types included focal and generalized seizures as well as febrile seizures. Continuous spikes and waves during sleep (CSWS) were reported in 8/13 individuals [13,16,23,34,37,38]. While both our cases with p.(Pro405Leu) show sleep-activated spikes, CSWS was incomplete or unilateral, reflecting a milder epilepsy syndrome and suggesting interindividual variability.

We identified the variant p.(Arg297Gln) in a 9-year-old male patient (#5) with infantile onset epileptic encephalopathy and ataxia, showing significant overlap to previously reported individuals with this GOF variant, who constantly developed ataxia [3,13,14,16,24,25,26,27,28,29].

In twins, one boy and one girl (#6 and #7), we identified the recurrent variant c.881G>A; p.(Arg294His), which was described previously in families with hereditary spastic paraplegia [15,20]. Similar to family 3 from Helbig et al., the presenting symptom in both children was epilepsy with different focal and generalized seizure patterns, including absences and status epilepticus. While previous reports showed spastic paraplegia (6/7 individuals) and ataxia (4/7 individuals) as prominent manifestations, we here present two additional cases with seizures that included tonic–clonic and absence seizures in both children [15]. Interestingly, ataxia was present in the boy only, and ataxia was markedly worsened when valproate was temporarily applied, resembling absence epilepsy with ataxia (AEA) due to loss-of-function mutations in *CACNA1A* [54]. Similar to previous AEA families, intrafamilial variability was present, as the girl showed isolated seizures with age-appropriate cognitive development, the boy had a severe complex developmental disorder, and the father did not show neurologic symptoms. The further family history revealed a sister of the unaffected father who had intellectual disability and an unspecified movement disorder. The mother of the father (grandmother of the reported twins) had a progressive gait disorder from advanced age resembling spastic paraplegia. Unfortunately, genetic testing was not available.

### 3.2. Functional Classification and Phenotypic Spectrum of Published KCNA2 Variants

Previously, three distinct subgroups of *KCNA2* variants with a genotype–phenotype correlation based on the electrophysiological channel properties were suggested [16]. We here analyze all available clinical reports and accordingly are able to group cases into these functional subgroups (LOF, GOF, and LOF/GOF). We show significant differences between subgroups regarding development before seizure onset, age at seizure onset, seizure outcome (seizure freedom), and intellectual disability. The combined dysfunctional effects of LOF/GOF variants are leading to more severe phenotypes compared to simple LOF or GOF variants. However, isolated LOF or GOF effects on K_v_1.2 both lead to epileptic encephalopathy as well. Recently, Shore et al. (2020) provided data, showing how GOF variants can lead to epilepsy [55]. By expressing known epilepsy-causing *KCNT1*-gain-of-function variants in excitatory and inhibitory neurons, they were able to demonstrate specific mutational effects affecting inhibitory neurons only, especially interneurons with fast-spiking activity, promoting network hyperexcitability and hypersynchronicity. Cell-type-specific functional changes from distinct variants might also explain the more severe epilepsy course in children carrying variants with mixed LOF and GOF effects. It can be hypothesized that specific effects dominate in different neuronal subpopulations, leading to an impaired GABA-ergic response from interneurons on one side and to impaired glutamatergic transmission in excitatory neurons on the other side, affecting different inhibitory and excitatory networks. Interestingly, the homologous *shaker* mutation of the LOF variant p.(Arg294His) was shown to lead to proton currents in addition to the pure loss-of-K_v_1.2 function when expressed in Xenopus laevis oocytes [15,56]. These effects are difficult to assess by most in vitro models, but in vivo neuron-subtype-specific dysfunctional effects with a loss of net potassium currents and additional a gain-of function from leaky proton currents might explain that the children described here exhibit focal and generalized absence seizures, which were previously attributed to gain-of-function effects in K_v_1.2.

Apart from the electrophysiological classification of variants, which allows assigning some phenotypic features, recurrent variants in each functional subgroup show distinct variant-specific characteristics. Individuals with the variant p.(Pro405Leu) are more likely to have febrile seizures and CSWS than other individuals in the LOF subgroup. Individuals with the variant p.(Arg294His) developed spastic paraplegia and ataxia as prominent manifestations. Seizures were described only in two previously reported cases [15] and in the two individuals reported here. Five individuals with the variant p.(Met255_Ile257del) presented with episodic ataxia, normal intellectual abilities and self-limited epilepsy, forming another distinct phenotype. In the GOF subgroup, apart from two individuals, all carried the recurrent variant p.(Arg297Gln). Therefore, characteristics of this subgroup are largely based on this variant. In the LOF/GOF subgroup, all but one individual with the variant p.(Thr374Ala) (*n* = 7) had neonatal refractory epilepsy. All showed severe intellectual disability, and in four of seven individuals, spastic cerebral palsy was reported (see Appendix A: Summary of clinical data).

### 3.3. Prediction of Functional Relevance and Modelling of KCNA2 Variants

By combining the distribution of known pathogenic and benign variants with prediction scores for all possible missense variants, we further define protein regions with enrichment of pathogenic variants and highly predicted mutational effects, such as the voltage-sensing domain, the pore-loop, and the pore-forming domains. Both novel variants described here are affecting residues within these regions. Homology modeling of p.(Pro407Ala) and p.(Pro417Tyr) suggests effects on protein stability and protein structure of K_v_1.2 (Figure 1). Importantly, we here complement the phenotypes associated with variants in both prolines of the PVP motif that similar to *KCNA1* are linked to severe early onset developmental and epileptic encephalopathy [52].

## 4. Materials and Methods

### 4.1. Database Research

Reports and entries of *KCNA2* variants containing clinical and/or genetic data were identified using PubMed, Human Gene Mutation Database (HGMD), and ClinVar (Figure 4). PubMed literature research was performed using the keywords “*KCNA2*” and “voltage-gated potassium channel K_v_1.2”. ClinVar database entries were selected for variants classified as “pathogenic” or “likely pathogenic”. Reported variants were checked for duplicate mentions. Data acquisition was finished 30 April 2020. In addition, we collected clinical and genetic data from seven previously unpublished cases from European epilepsy centers with putatively causative variants in *KCNA2*. After the exclusion of one individual with a paternally inherited variant of uncertain significance (VUS) and all individuals without sufficient clinical information (37 ClinVar, 8 HGMD/Pubmed), 76 individuals were included into further statistical analyses (Figure 4).

Individuals were classified according to functional consequences of the mutated K_v_1.2 channel when functional data from in vitro analysis were available. Quantitative variables are illustrated by sample size, mean, and standard deviation. Statistical comparisons between means of two cohorts were performed with Welch’s two-sample t-test. Frequencies of descriptive variables are depicted with number, sample size, and percentages. Statistical comparisons of the frequency of descriptive variables between cohorts were conducted using the two-tailed Fisher’s exact test.

### 4.2. Computational Pathogenicity Analysis and 3D Protein Structure Modeling

To visualize differences in variant distribution between affected individuals and unaffected controls, we compiled all currently known pathogenic and benign *KCNA2* variants and their respective allele count using the databases PubMed, HGMD, ClinVar, and gnomAD. gnomAD variants referring to the canonical Ensembl transcript ENST00000485317.1 were selected by the categories “Missense” and “pLOF” (excluding frameshift variants). Variants were standardized to the NM_004974.3 transcript of the GRCh37/hg19 human reference genome using the Mutalyzer Nomenclature Checker web tool [57]. Additionally, all biologically possible missense variants of the *KCNA2* coding sequence were computed and translated to the corresponding genomic position. Subsequently, established variant effect prediction (VEP) scores (REVEL, CADD, MetaLR, MetaSVM, M-CAP, PolyPhen-2, SIFT) were annotated using the VEP web tool of Ensembl Release 101 (http://grch37.ensembl.org/Tools/VEP, accessed on 18 January 2021; versions: dbNSFP v. 4.1a, REVEL v. 3.5a, CADD v. 1.6, M-CAP v. 1.3, PolyPhen-2 v. 2.2.2, SIFT v. 5.2.2) in order to evaluate the likelihood of pathogenic variant effects [58,59,60,61,62,63,64,65]. Resulting VEP scores of the canonical Ensembl feature and variant distributions were plotted along the primary structure of the K_v_1.2 protein using the *geom_smooth* and *geom_density* function, respectively, of the *ggplot2* library in RStudio (v. 1.2.5042; RStudio, Inc., Boston, MA, USA) [66]. The REVEL meta-score demonstrated high overall performance regarding discrimination between pathogenic and benign single nucleotide variants in large clinical variant datasets [43]. Therefore, the REVEL score is primarily used in this study to assess the pathogenicity of variants. A complementary synopsis of all annotated VEP score results is depicted in Appendix A.

Localization and structural consequence of the identified *KCNA2* missense variants p.(Pro405Leu), p.(Pro407Ala), and p.(Tyr417Cys) in the quaternary protein structure were analyzed and visualized using PyMol (v. 2.5.0a; Schrödinger, LLC, New York, NY, USA) installed through Anaconda (v. 2020.11 with Python 3.8.5; Anaconda Inc., New York, NY, USA) and the existing PDB template 3LUT (chain B: α-subunit) of rat K_v_1.2.

## 5. Limitations

Due to limited and heterogeneous quality of data in publications, phenotypic features might not have been completely accessible. Therefore, the number of patients with available information is specified with each clinical feature. Due to differing age at imaging, the detectability of cerebellar changes might vary between reports. Despite the reasonable ability of the software programs to predict pathogenicity, they still cannot discriminate among functional LOF, GOF, and LOF/GOF-type variants. This is a critical limitation of computed predictions of variant effects in clinical practice and for genetic counseling. Furthermore, some recurrent variants represent a significant proportion of the cohort and variant-specific effects may dominate the phenotypic presentation of the whole functional subgroup.

## 6. Conclusions

Our study provides further insights into the clinical spectrum, genotype–phenotype correlations, variability, and predicted functional impact of *KCNA2* variants. The location of variants reported here highlights the importance of the PVP motif in transmembrane domain S6 as a mutational hotspot. A novel case of self-limited infantile seizures suggests a continuous clinical spectrum of *KCNA2*-related disorders. The prediction of functional relevance of variants allowed us to identify hotspots of functional impact and recurrent mutations.

## Figures and Tables

**Figure 1 ijms-22-02824-f001:**
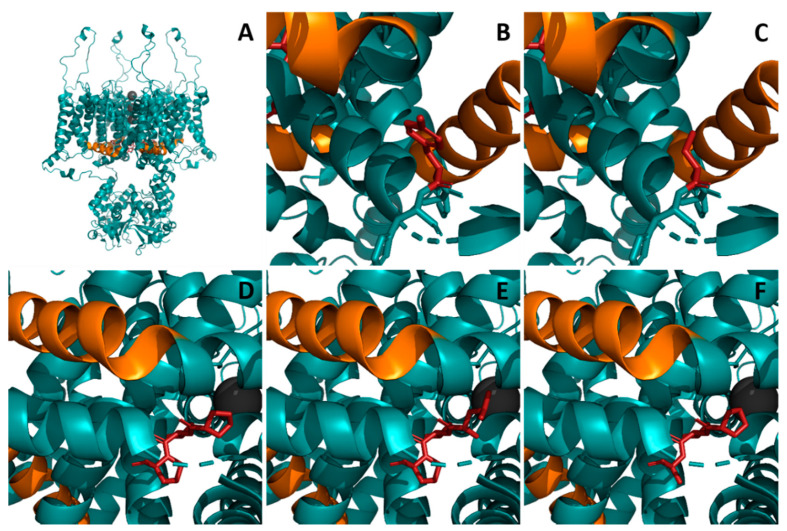
Localization of three identified missense variants in the K_v_1.2 channel. Three-dimensional (3D) protein structure of the K_v_1.2 channel shown as a ribbon model. (**A**) Lateral overview of the channel tetramer (turquoise) and three potassium ions localized in the channel pore (spheres, gray). The residues Pro405, Pro407, and Tyr417 are shown in stick representation (red). The S4-S5 linker is highlighted in orange. (**B**) Close-up view of residue Tyr417 and (**C**) the mutated residue Cys417 (red sticks), in context with the S4-S5 linker regions of the same and neighboring channel subunits, respectively (orange). (**D**–**F**) Close-up views of the proline(405)–valine(406)–proline(407) (PVP) motif (red sticks; (**D**)) as well as the amino acid substitutions Leu405 (**E**) and Ala407 (**F**) of the corresponding proline residues. The PVP motif interacts with the S4–S5 linker, and both prolines of this symmetric motif (**D**) are important for channel gating. The substitution of Pro405 (**E**) and Pro407 (**F**) disrupts its symmetry and structure, likely impairing interactions with S4–S5 during gating-related conformational changes. All three substitutions are located in the S6 helix and in proximity to the S4–S5 linkers, which is suggestive for the probable affection of associated interactions leading to altered channel gating.

**Figure 2 ijms-22-02824-f002:**
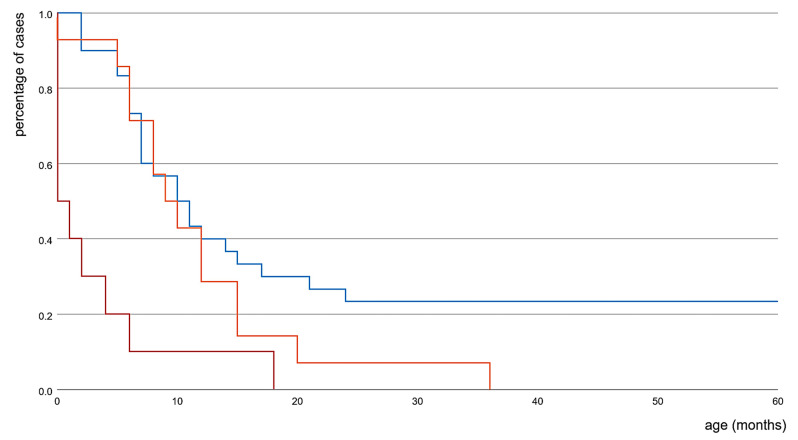
Time-to-Event Curve. Percentage and age of first seizures of the functional subgroups LOF (blue), GOF (orange), and LOF/GOF (red).

**Figure 3 ijms-22-02824-f003:**
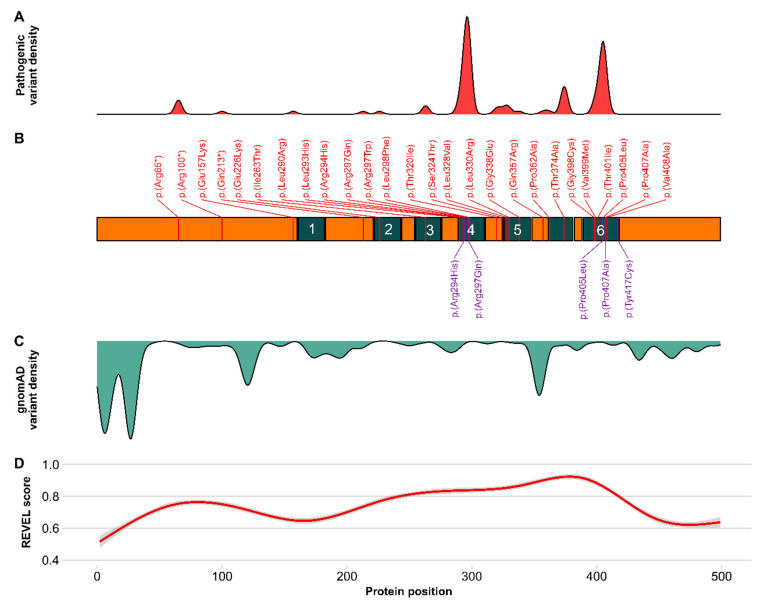
Distribution of pathogenic and benign (tolerated) *KCNA2* variants along the primary structure. (**A**) Density of pathogenic variants reported in Pubmed, HGMD, and ClinVar (red). (**B**) Linearized KCNA2 protein model including cytoplasmic (orange) as well as transmembrane and pore (green) domains. The amino acid position and description of previously published variants and variants reported here are highlighted in red (top) and violet (bottom), respectively. (**C**) Density of tolerated variants reported in gnomAD (light green). (**D**) Polynomial regression model of the REVEL ensemble pathogenicity score for all possible missense variants according to their position in the linearized K_v_1.2 protein. Enrichment of disease-associated variants can be observed in the transmembrane domains S4–6 and the pore loop. Distribution of pathogenic compared to benign variants is mainly reciprocal. REVEL scores approximately recapitulate pathogenic variant distribution.

**Figure 4 ijms-22-02824-f004:**
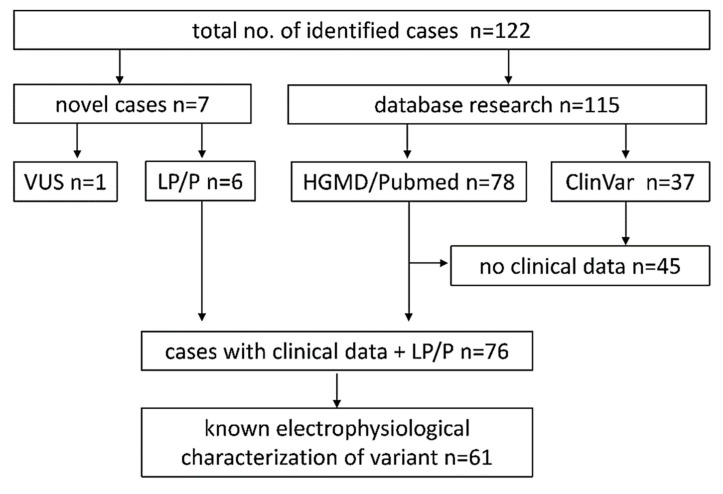
Flow-chart of *KCNA2* cases identified through database research and collection of unpublished cases. VUS: variant of uncertain significance, LP/P: likely pathogenic/pathogenic, HGMD: Human Gene Mutation Database, PubMed: public medicine database of the National Institutes of Health, ClinVar: Clinically Relevant Sequence Variations database.

**Table 2 ijms-22-02824-t002:** Variants of *KCNA2* reported in ClinVar.

cDNA Change	Protein Change	Interpretation	Phenotype	Number of Reported Patients
**missense**				
c.788T > C	p.(Ile263Thr)	LP	epileptic encephalopathy	2
c.869T > C	p.(Leu290Pro)	LP	epileptic encephalopathy	1
c.881G > A	p.(Arg294His)	LP	epileptic encephalopathy	4
c.890G > A	p.(Arg297Gln)	P	epileptic encephalopathy	6
c.894G > T	p.(Leu298Phe)	P	epileptic encephalopathy	2
c.959C > T	p.(Thr320Ile)	P	epileptic encephalopathy	1
c.989T > G	p.(Leu330Arg)	LP	not provided	1
c.1013G > A	p.(Gly338Glu)	LP	epileptic encephalopathy	1
c.1084C > G	p.(Pro362Ala)	LP	not provided	1
c.1120A > G	p.(Thr374Ala)	P	epileptic encephalopathy	2
c.1195G > A	p.(Val399Met)	LP	epileptic encephalopathy	2
c.1202C > T	p.(Thr401Ile)	LP	not provided	1
c.1214C > T	p.(Pro405Leu)	P	epileptic encephalopathy	8
c.1219C > G	p.(Pro407Ala)	LP	not provided	2
c.1223T > C	p.(Val408Ala)	P	epileptic encephalopathy	1
**truncating**				
c.298C > T	p.(Arg100*)	LP	epileptic encephalopathy	1
**deletion**				
g.(?_110593873)_(110604802_?)del		P	epileptic encephalopathy	1

all variants are described using the transcript NM_004974.3; LP: likely pathogenic; P: pathogenic.

**Table 3 ijms-22-02824-t003:** Genetic and clinical characteristics of novel cases with *KCNA2* variants.

Case	1	2	3	4	5	6	7
cDNA Change	c.1250A > G	c.1219C > G	c.1214C > T	c.1214C > T	c.890G > A	c.881G > A	c.881G > A
Protein Change	p.(Tyr417Cys)	p.(Pro407Ala)	p.(Pro405Leu)	p.(Pro405Leu)	p.(Arg297Gln)	p.(Arg294His)	p.(Arg294His)
Inheritance	parental	de novo	de novo	de novo	de novo	Familial (sister of #7)	familial (brother of #6)
Age at inclusion/Gender	3 y/m	3 y/f	8 y/f	2 y/m	9 y/m	8 y/f	8 y/m
Development before seizure onset	normal	na	normal	normal	normal	mildly delayed motor development	delayed speech and motor development
Epilepsy onset (months)	7.5	0.25	21	7	8	72	24
Seizure types	bilateral tonic–clonic	generalized onset tonic and hypomotor seizures with cyanosis	prolonged febrile motor seizure	prolonged focal motor onset seizures and bilateral tonic–clonic seizures, febrile seizures, postictal hemiplegia	generalized onset bilateral tonic–clonic, atypical absences from 2y3m	febrile seizures; generalized onset myoclonic and absence seizures, status epilepticus	febrile seizures; focal and generalized onset, absence seizures, myoclonic and tonic seizures, status epilepticus
Seizure outcome	seizure free at 10 months	seizure-free at 2 years	ongoing	seizure free at 2 years	ongoing	ongoing	ongoing
ID	normal	severe ID	moderate ID	normal	learning disability	mild ID	moderate ID
Movement disorders	no	no	no	episodic ataxia	ataxia	mild tremor	ataxia, tremor, worsening on VPA
EEG features	normal at 3y	multifocal spikes and reduced physiologic sleep characteristics	multifocal spikes in both centro-temporal regions, evolving to hemi-CSWS during follow-up	multifocal spikes, activation during sleep	na	multifocal spikes and polyspikes frontal photoparoxysmal reaction, 3–4 Hz spike waves	generalized slowing, activation of epileptiform activity by photostimulation, multifocal spikes, 3–4 Hz spike waves
Magnetic resonance imaging	normal	normal	normal	left sided hippocampal sclerosis	normal	cystic lesion subcortical frontal left DD DNET	cerebellar atrophy (vermis)
Additional findings		SGA/dystrophy	dysarthria	polydactyly left	dysarthria and stuttering	muscular hypotonia	dystrophy, microcephaly, muscular hypotonia
Comments					reported in ClinVar	reported in ClinVar	

ID: intellectual disability, n/a: not available; CSWS: continuous spikes and waves during sleep. CSWS was defined by a Spike-Wave-Index (SWI) of at least 85% in an epoch of at least 10 min duration after onset of alpha attenuation or clinical signs of sleep; DNET: dysembryoblastic neuroectodermal tumor; SGA: small for gestational age; na: not available; f: female; m: male; y: years.

**Table 4 ijms-22-02824-t004:** Phenotypic features of reported and novel patients, grouped according to known functional consequences on K_v_1.2.

*n* = 61	LOF (*n* = 35)	GOF (*n* = 16)	GOF/LOF (*n* = 10)
Variants	Pro405Leu (16)	Arg297Gln (14)	Thr374Ala (7)
	Arg294His (9)	Glu157Lys (1)	Leu290Arg (1)
	Met255_Ile257del (7)	Leu298Phe (1)	Leu293His (1)
	Gln213 * (1)		Leu328Val (1)
	Ile263Thr (1)		
	Gly398Cys (1)		
Development before seizure onset	normal (29/34)	normal (9/12)	normal (3/8)
	impaired (5/34)	impaired (3/12)	impaired (5/8)
Epilepsy onset age mean/median (SD) (months)	early childhood 17.4/8.75 (SD 31.2) (1 outlier with 156 m)	infantile or early childhood 11.6/9.5 (SD 8.6)	neonatal or early infantile 3.1/0.5 (SD 5.6) (1 outlier with 18 m)
Seizure types	focal only (5/30)	focal only (0/13)	focal only (5/10)
	generalized only (14/30)	generalized only (9/13)	generalized only (3/10)
	both (11/30)	both (4/13)	both (2/10)
Febrile seizures	yes (13/27)	yes (5/12)	yes (1/9)
	no (14/27)	no (7/12)	no (8/9)
Seizure outcome	never seizures (5/32) **		
	seizure-free (13/32)	seizure-free (2/13)	seizure-free (0/9)
	uncontrolled (14/32)	uncontrolled (11/13)	uncontrolled (9/9)
Intellectual disability	normal (9/35) ***	normal (0/14)	normal (0/10)
	mild (9/35)	mild (3/14)	mild (1/10)
	moderate (12/35)	moderate (7/14)	moderate (0/10)
	severe (5/35)	severe (4/14)	severe (9/10)
Movement disorders	ataxia (18/28)	ataxia (16/16)	ataxia (4/4)
	tremor (7/28)	tremor (7/16)	tremor (1/5)
	spasticity (6/28) **	spasticity (0/16)	spasticity (4/7)
	episodic ataxia (6/28) ***		
CSWS	(9/20)	(0/11)	(0/7)
cMRI	normal (20/26)	normal (5/14)	normal (5/10) ^##^
	cerebellar atrophy (2/26)	cerebellar atrophy (8/14)	cerebellar atrophy (5/10)
	hippocampal sclerosis (1/26)	subcortical white matter lesions (1/14)	
	DNET (1/26), other (2/26)		
Additional findings	- short stature and growth hormone deficiency, subclinical hypothyroidism (Syrbe et al., P4)	- scoliosis, kyphosis, genu valgum (Masnada et al., P12)	- facial dysmorphism (Masnada et al., P20)
	- episodic ataxia (Corbett et al.)	- facial dysmorphismes (Syrbe et al., P2)	- bilateral optic atrophy (Masnada et al., P22,23)
	- hereditary spastic paraplegia (Helbig at al., P1–5; Manole et al.)	- small nose and mouth, hepatic lesion of unknown origin (Masnada et al., P15)	- microcephaly
	- autism spectrum disorder (Helbig et al., P2)	- autism spectrum disorder (Ngo)	

The number of cases with specific clinical features is stated in relation to number of cases with available information concerning this feature. Responsive to treatment was defined as reported significant reduction of seizures. CSWS: continuous spikes and waves during slow sleep. DNET: dysembryoplastic neuroectodermal tumor. P: patient, m: months; ** p.(Arg294His); *** p.(Met255_Ile257del); ## due to neonatal epilepsy onset, cMRI was performed in first months of life.

## Data Availability

The data presented in this study are available on request from the corresponding author.

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
