# Peer review of "Refining Genotypes and Phenotypes in KCNA2-Related Neurological Disorders"

_ijms, 2021, doi:10.3390/ijms22062824_

Round 1
Reviewer 1 Report
Variants in the KCNA2 gene encoding for Kv1.2 voltage-gated potassium channel subunits have been recently described as associated to human phenotypes ranging from self-limiting familial seizures to severe developmental encephalopathies and intellectual disabilities, in addition to movement disorders. The present paper by Doring et al. describes seven additional individuals harboring three known and two novel KCNA2 variants, and also provides a critical review of all cases described in the literature and in genomic databases for both potentially pathogenic (Clinvar) and non-pathogenic (GnomAD) variants. The novel data support provide further insight into the phenotypic spectrum associated to KCNA2 variants, and analyze available data according to useful prediction genotype-phenotype correlations emerging for this gene, with GOF/LOF variants tools (REVEL).
I have a few comments needing to be addressed:
- In the Introduction (lines 74-77) the sentence is misleading, and leads the reader to erroneously believe that there is a single ”voltage-gated potassium channel complex”. Please, rewrite clarifying that the genes mentioned encode for proteins potentially associated to or modulating Kv1.x channel function.
- At line 147, it is not clear why the authors include 76 out of the 115 cases described in the literature. Please clarify.
- In the Results section (Table 3), the definition of CSWS should be given, as this EEG trait might be prone to misinterpretation. Maybe a better description should be given (see also Sands et al., Ann Neurol. 201; https://doi.org/10.1002/ana.25522); sleep‐activated multifocal epileptiform discharges is an alternative and maybe more appropriate definition, whenever sleep EEG data at an appropriate age are available.
- I have had hard times in interpreting Fig. 3D. The Authors should specify better why REVEL score was chosen to be explicitly shown and why this score gives is endowed with an improved predictive power when compared to other prediction tools. I don’t mean (in this context) a rigorous statistical comparison, but a better description of the tool and of its advantages over others should be given to allow the reader to follow the manuscript. Also, some of the variants reported in the top part of Fig. 3B are not listed in Table 1 (p.Arg100*, for example); could they explain why this is the case?
- Also, in the study limitations I would add that, despite the reasonable ability of the softwares to predict pathogenicity, they still cannot discriminate among LOF/GOF/LOF-GOF-type variants; this is a critical limitation if such prediction is to be implemented in real clinical practice and for genetic counseling.
Reviewer 2 Report
Döring et al., conduct a review of the clinical phenotype and a description of the functional consequences associated with pathogenic variants in KCNA2, both in reported and novel cases. As a consequence, they delineate phenotypic as well as functional subgroups of KCNA2-associated disorders.
The article requires a minor revision:
- Title (line 112) must appear after the description of Figure 1, and not as a title.
- Table 3. Dysarthria and muscular hypotonia are not movement disorders. These symptoms can be included in additional findings in the same table.
- Line 191. Spasticity is not a movement disorder either. Do you mean spastic paraparesis? Please correct this term accordingly.
- In this article, the description of all associated symptoms and signs (ataxia, tremor, cerebellar atrophy, etc.) is rather simple. It is the mere depiction in percentage of the symptom. From my perspective, after reviewing the literature, you should be able to delve more deeply into the characteristics of each of these symptoms (age at onset, location, other characteristics). For example, in line 197, can you comment on the age at which cerebellar atrophy is observed? What areas of the cerebellum does it affect? is it generalized or affects the vermis more than the hemispheres?
- On the other hand, can you prepare a better representation of the analyzed data. For example, can you express the age at onset of epilepsy of Table 4 with a Kaplan-Meir curve? This graph will help us all to understand better the differences between the three groups.
- In Table 1, you describe a significant number of patients harboring c.890G>A and c.1214C>T variants. There are patients with homozygous variants c.193C>T. Do these patients have special characteristics, are they more severe, etc.? Have you attempt to establish a genotype-phenotype characterization? If you have done it and you have not found anything, it is important to say so.
- In the Tables 1 and 2 you make a differentiation between Epileptic encephalopathy, early infantile, 32 and epileptic encephalopathy. You should unify to the term Epileptic encephalopathy, early infantile, 32 throughout the article.
- There is some confusion in the way you present certain data. For example, in Line 263 there are specific data that should appear in the results section but instead they are only mentioned for the first time in the discussion section.
- The discussion section must be improved. Your work is very good and I think you can contribute more to the discussion section.
- A supplementary table including all the clinical data from the literature review should be included.
Reviewer 3 Report
The main objective of this article is to collect all previously described and published mutations in KCNA2 that are associated with a pathology. In addition, some mutations that had not been described previously are included.
The authors, first, group the different mutations described in relation to the functional consequence of said mutation in the potassium channel. There are mutations that are associated with a gain of function (GOF), others with a loss of function (LOF), and others with mixed effects. Moreover, comparing phenotypic characteristics from patients with functional data on mutational effects, they found that carriers of variants leading to complex and mixed channel dysfunction that are associated with a gain- and loss-of-potassium conductance more often show early developmental abnormalities and an earlier onset of epilepsy compared to patients with pure loss-of-function or pure gain-of-function variants, respectively. However, authors did not discuss this important point and they did not give a possible explanation for that.
Authors should introduce a possible explanation for this important finding in the discussion, as well as how both the gain and loss of channel function translate into epileptic encephalopathy.
